# Developing engaged and 'teamful' leaders: A randomized controlled trial of the 5R identity leadership program

S. Alexander Haslam[1]*, Jordan Reutas[1], Sarah V. Bentley[1], Blake McMillan[1], Madison Lindfield[1], Mischel Luong[1], Kim Peters[2], Niklas K. Steffens[1]

1 School of Psychology, University of Queensland, Brisbane, Australia, 2 Exeter Business School, University of Exeter, Exeter, United Kingdom

* a.haslam@uq.edu.au

**Data Availability Statement:** All relevant data for this study are publicly available in the OSF repository (https://osf.io/e82bd/).

## Abstract

The social identity approach to leadership argues that leaders' capacity to influence and inspire others is grounded in a shared sense of social identity (or 'us-ness') that those leaders create, advance, represent, and embed for the groups they lead. The approach therefore argues that a key task for leaders is to develop insights and skills of (social) identity leadership that allow them to motivate and mobilize groups and transform them into a potent social and organizational force. In contrast to other approaches and programs which focus on leaders' leader identity (their 'I-ness'), the 5R leadership development program supports the development of leaders' social identity by raising awareness of the importance of social identity ('we-ness') for leadership and taking leaders through structured activities that help them build engaged and inclusive teams. The present research assessed the benefits of facilitated and learner self-directed versions of the 5R program ($N$s = 27, 22 respectively) relative to a no-treatment control ($N$ = 27). Results (including those of an intention-to-treat analysis; $N$ = 76) indicated that, relative to leaders in the control condition, those who participated in both forms of 5R reported large increases in identity leadership knowledge, as well as medium-sized increases in both team engagement (a compound factor comprised of team identification, team OCB, team efficacy, and work engagement) and 'teamfulness' (comprised of team reflexivity, team psychological safety, team goal clarity, and inclusive team climate). We reflect on the importance of teamfulness for leadership and team functioning and on the value of programs that help leaders develop this.

## Introduction

The social identity approach to leadership argues that leaders' capacity to influence and inspire others is grounded in a shared sense of social identity (or 'us-ness') that those leaders create, advance, represent, and embed for the groups they lead [1–4]. The approach therefore argues that a key task for leaders is to acquire insights and skills of (social) *identity leadership* that allow them to motivate and mobilize the groups they are seeking to lead in order to transform those groups into a potent organizational and social force. This approach differs from

**Funding:** The author(s) received no specific funding for this work.

**Competing interests:** I have read the journal's policy and the authors of this manuscript have the following competing interests: Intellectual property pertaining to the 5R Leadership Development Program is asserted, owned, and licensed by the University of Queensland. The university and its agents have a commercial as well as an intellectual interest in this program. The fourth author is the Director of Identify Leadership which is formally licensed to deliver 5R. This does not alter our adherence to PLOS ONE policies on sharing data and materials.

prevailing approaches to leader development which focus largely on improving the skills and mindsets of individual leaders (for reviews, see [5, 6])—with work which has evaluated such programs indicating that they generally succeed in developing a person's *leader identity* (their sense of themselves as 'me the leader' [7–10]). In contrast to this focus on 'me' and 'I', the social identity approach embraces the idea that leadership—and hence leadership development—is fundamentally a group process that requires leaders to *look outwards* towards their teams and to develop sense of 'we' and 'us' [11, 12].

The social identity approach to leadership has become increasingly influential over the course of the last decade (see [11] for evidence). There are two key reasons for this. The first is that a burgeoning body of research supports the key tenets of the approach. For example, a recent meta-analysis of 128 studies found that leadership is more effective in shaping the behavior of followers to the extent that leaders are perceived to be *prototypical* (i.e., representative) of a social identity that they share with those followers (i.e., seen as 'one of us'; [13, 14]). Importantly, as well as showing that leaders' identity prototypicality predicts their behavioral impact (i.e., whether their leadership translated into others' *followership*; [12]), this meta-analytic evidence indicates that this relationship was also present in experimental studies that *manipulated* leaders' identity prototypicality and therefore established the causal status of this as a determinant of leader effectiveness.

A second reason for the impact of the social identity approach is that the process of investigating and garnering evidence around it has led to the creation of a large community of researchers interested in exploring both the dynamics of identity leadership and their relevance for organizational and social outcomes. In particular, the Global Identity Leadership Develop (GILD) project has brought together more than 50 researchers from over 25 countries to investigate the cross-cultural dimensions of identity leadership and the robustness of its contribution to multiple aspects of organizational functioning [15]. As well as confirming that leaders' identity prototypicality is a reliable predictor of their influence, work on this project has shown that prototypicality is also associated with team members' job satisfaction and team identification. Beyond this, though, the GILD project shows that other components of identity leadership make important and distinctive contributions to group vitality. For example, leaders' work as *identity champions* (who 'do it for us' [16]) has been found to contribute to employees' job satisfaction, while their work as both *identity entrepreneurs* (who 'create a sense of us' [17]) and *identity impresarios* (who 'make us matter' [1]) has been linked to greater team cohesion and engagement.

Research in a number of other fields (notably sport and politics) underscores the importance of these different components of identity leadership for both leader effectiveness and group success (for reviews see [18, 19]). Amongst other things, this is because leaders who cultivate a sense of shared social identity in their teams help to create a sense of trust, psychological safety, and collective confidence [20, 21], while also encouraging collective effort [22, 23]. On top of this, the most recent work on the GILD project has shown that identity leadership helps to drive team members' creativity and innovation [24] while also supporting their mental health and well-being [25–27].

## Developing identity leadership

In light of this accumulating evidence, two obvious questions that arise are whether and how would-be leaders can be helped to develop the insights and skills of identity leadership. These are also questions that have been of increasing interest to researchers and practitioners in recent years. Early efforts to provide answers led to the development of an evidence-based framework for leaders to *Actualize Social and Personal Identity Resources* within the groups

and teams they lead—the ASPIRe model [28, 29]. In particular, this model grapples with challenges of diversity management and strategic planning of a form encountered across a wide range of organizational contexts (e.g., as discussed by [30–34]). It then seeks to tackle these challenges by taking leaders and their groups through a structured program of activities in which they identify and work with diverse organizational identities (e.g., as members of distinct organizational units) before seeking to integrate these identities within an organic superordinate identity.

Studies in an array of organizational and other social settings have confirmed that the ASPIRe model provides a viable framework for practitioners to work with leaders in order to harness the power of social identities [35–37]. In particular, they provide evidence that leaders who want to manage groups effectively need to attend to what Haslam and colleagues (2011) refer to as the "3 Rs" of identity leadership. The first of these, *Reflecting*, involves taking stock of the nature of the social identities that are important for members of a given organization (notably through a process of social identity mapping [38–40]). Second, *Representing* involves clarifying the content and meaning of both diverse and shared identities (notably though subgroup caucusing; [36, 40]). Then, third, *Realizing* involves working with group members to help them achieve their various identity-related ambitions (notably through goal-setting and initiation of group-supporting structure [41, 42]).

Although the ASPIRe model was concerned with matters of leadership, it was designed to address a relatively narrow set of challenges that leaders face rather than to develop identity leadership more broadly. To give the model broader relevance, Haslam and colleagues [43] therefore set about translating its core components into a program explicitly designed to support (identity) leadership development. This centered on three workshops that focus on the 3Rs of identity leadership and which guide leaders through activities (e.g., social identity mapping, participative collective goal setting) that they are subsequently required to conduct with their team members with a view to developing a 'hands on' appreciation of identity management (see also [44]). These were also contextualized by two additional workshops: an initial *Readying* session that informs participants about the importance of group and social identity processes for leadership, and a concluding *Reporting* session in which leaders reflect on progress towards collective goals and on learnings associated with the program as a whole. The program thus has five core modules, and it is from these that it derives its name—5R.

As originally conceived, 5R incorporates a number of distinctive features that accord with what systematic review and meta-analysis reveal to be best practice in the field [45, 46]. In particular, it (a) is conducted face-to-face, (b) is demanding for participants, (c) involves action not just reflection, (d) is conducted in the contexts where leaders actually operate (i.e., on-site rather than off-site), (e) has a spaced sequence of activities with clear purpose, and, most critically, (f) involves engagement with those who are actually being led [46–50]. In this way, the 5R program is designed to include, connect with, and mobilize the teams for which leaders have responsibility rather than to exclude those teams from the leadership process and the broader dynamics of organizational development and change.

Initial trials of 5R have provided evidence of its coherence, viability, and effectiveness as a leadership development program. In the first of these, Haslam et al. [43] took a diverse group of allied health leaders through the program and found that participation increased participants' self-reported ability to engage in identity leadership, as measured by the *Identity Leadership Inventory* (ILI) which captures the four core components of identity leaderships [3]. Importantly too, the program increased leaders' experience of team goal clarity as well as their team identification. Moreover, regression analysis indicated a dose-response relationship, whereby this uplift was more pronounced the more program activities participants took part in and the more they reported engaging with the program's content. Interestingly, though,

while 5R increased participants' ability to engage in identity leadership, it reduced their motivation to develop a distinct identity as individual leaders (i.e., their *leader identity* [51–53]). This speaks to the program's success as a vehicle for engaging with the 'we' of leadership rather than the 'I' [11].

Other trials—many with leaders in the field of sport—have also produced encouraging results. In Britain, Slater and Barker [44] ran a variant of 5R with leaders of the national paralympic football team over a period of two years and found that it led to significant increases in social identification among those leaders, and also to increases in the degree to which athletes in the team felt that their leaders were displaying identity leadership (as assessed by the ILI). The program also led to an increase in the number of hours of practice that those athletes completed outside formal training camps.

Additional trials of versions of 5R have also been conducted in the Netherlands and Australia with business leaders as well as leaders of professional and semi-professional sporting teams. Qualitative data from these studies indicates that leaders and team members perceived the program to be an effective way of bringing people together around a sense of shared identity in ways that help the team move forward as a unit [54, 55]. Researchers have also developed a spin-off of 5R, 5RS, for use in a sporting contexts that combines 5R with an emphasis on shared leadership within teams [54]. An experimental study by Mertens et al. [56] with eight national-level basketball teams in Belgium showed that in comparison to a no-treatment control, participation in 5RS served to strengthen both leaders' and team members' identification with their team as well as all respondents' intrinsic motivation, commitment to team goals, and well-being.

## The present study

Notwithstanding growing evidence of the feasibility and efficacy of 5R as a vehicle for leadership development, there are three notable gaps in this evidence base. First, none of the trials that have been conducted in organizational contexts have had a randomized design that allows the program's impact to be established and gauged relative to a no-intervention comparison condition [57, 58]. Indeed, this is emblematic of a broader deficit in the organizational literature, which is attributable, at least in part, to the logistical and resource-related challenges of conducting randomized controlled trials in this domain [58–60].

Second, in all the quantitative studies that have been conducted to date, 5R has been delivered by one or more of the researchers responsible for developing the program. Because these researchers were committed to the program's success, this raises the possibility of experimenter bias [61], as well as the more general question of whether the benefits of 5R can be achieved when it is delivered by facilitators who are less invested in, and less knowledgeable about, the social identity approach.

Third and finally, as noted above, in line with recommendations in the literature (e.g., [46]), all previous trials of the 5R program have involved face-to-face delivery. However, this mode of delivery was not possible during the COVID-19 pandemic and, in its wake, the move to flexible working practices has also made this face-to-face delivery logistically more challenging [62]. This raises the question of whether the program could be adapted for more versatile on-line delivery, and if it were, whether this would undermine the program's potency in any way.

With these three issues in mind, the present study took the form of a randomized controlled trial of a new online-facilitated version of 5R delivered by a team of postgraduate organizational psychology students as part of their formal professional training. The most basic hypothesis that the study tested was that—relative to a no-treatment control—the program would

increase participants' knowledge and skills of identity leadership (H1). This was assessed using an adapted version of the ILI (following [3]).

As well as expecting 5R to have a positive impact on participants' ability to engage in identity leadership, in line with the logic of previous research, we also anticipated that it would have limited impact on participants' desire to develop and promote their *leader identity* [51–53]. Accordingly, following the procedure used by Haslam et al. [43], we also included a measure of participants' *leader identity pursuit* as a control variable that has superficial similarity to our target dependent variable (identity leadership knowledge), but which taps different theoretical processes (as discussed by [11]). In line with the logic of non-equivalent dependent variable design [63, 64] this allows us to rule out the possibility that any changes we observe arise from non-specific responses to the intervention (e.g., testing effects and common method variance [65, 66].

More substantively, we hypothesized that participation in the program would increase leaders' self-reported ability to work with their teams to address their collective challenges and goals (H2), and that this would be explained by the knowledge and skills of identity leadership that they had acquired (H3). H2 was assessed using a range of measures intended to capture various inter-related aspects of collective team functioning that have previously been shown to benefit from identity leadership and from the sense of shared social identity that it harnesses (e.g., team identification, team goal clarity, team reflexivity, team psychological safety [15, 21, 43]. H3 was assessed using regression-based mediation analysis.

However, in addition to this, the development of an on-line version of 5R allowed us to assess the value of program facilitation per se. We did this by comparing a *Facilitated* condition in which program modules were facilitated via on-line video conferencing with a *Self-Directed* condition in which engagement with the same content was unfacilitated. In line with previous findings (e.g., as reported by [46]), we hypothesized that participation in 5R would prove more beneficial when it was facilitated rather than self-directed. To explore this possibility, after (a) testing H1, H2 and H3 by comparing responses across treatment and control conditions, we (b) went on to compare responses across facilitated and self-directed conditions.

## Method

### Participants and design

The study was approved by the School of Psychology Ethics Review Committee at the first author's university (application number 2020/HE002261). The researchers distributed an invitation to participate in a new online version of the 5R leadership development program via a range of formal and informal networks. An a priori power analysis (G*Power, version 3.1.9.6) indicated that a sample of at least 52 participants would be required to give an 80% chance of detecting a moderate-to-strong effect similar in magnitude to that observed by Haslam et al. [43] (with 2-tailed $\alpha$ = .05). On the basis of previous 5R research, we assumed there would be an attrition rate of around 30% and so aimed to recruit 75 participants.

The invitation attracted interest from a wide range of respondents, of whom 76 met the criteria for inclusion—namely in being leaders of teams with at least two members. Of the 43 participants who completed the trial and provided T2 data (i.e., 57% of those recruited for the study), 25 were male, 18 were female. The majority (41%) were aged between 46 and 55, 31% were between 36 and 45, 17% were under 35 and 12% were over 55. Most (52%) described themselves as having a senior position, with the remainder (48%) describing their position as being of 'intermediate' seniority. Due to a high level of engagement via the head of leadership development in a large construction company (with six subsidiary companies) the majority of participants (around 70%) had leadership responsibility for teams working in diverse areas of construction and manufacturing. On average these teams had 5.91 members (*SD* = 0.50).

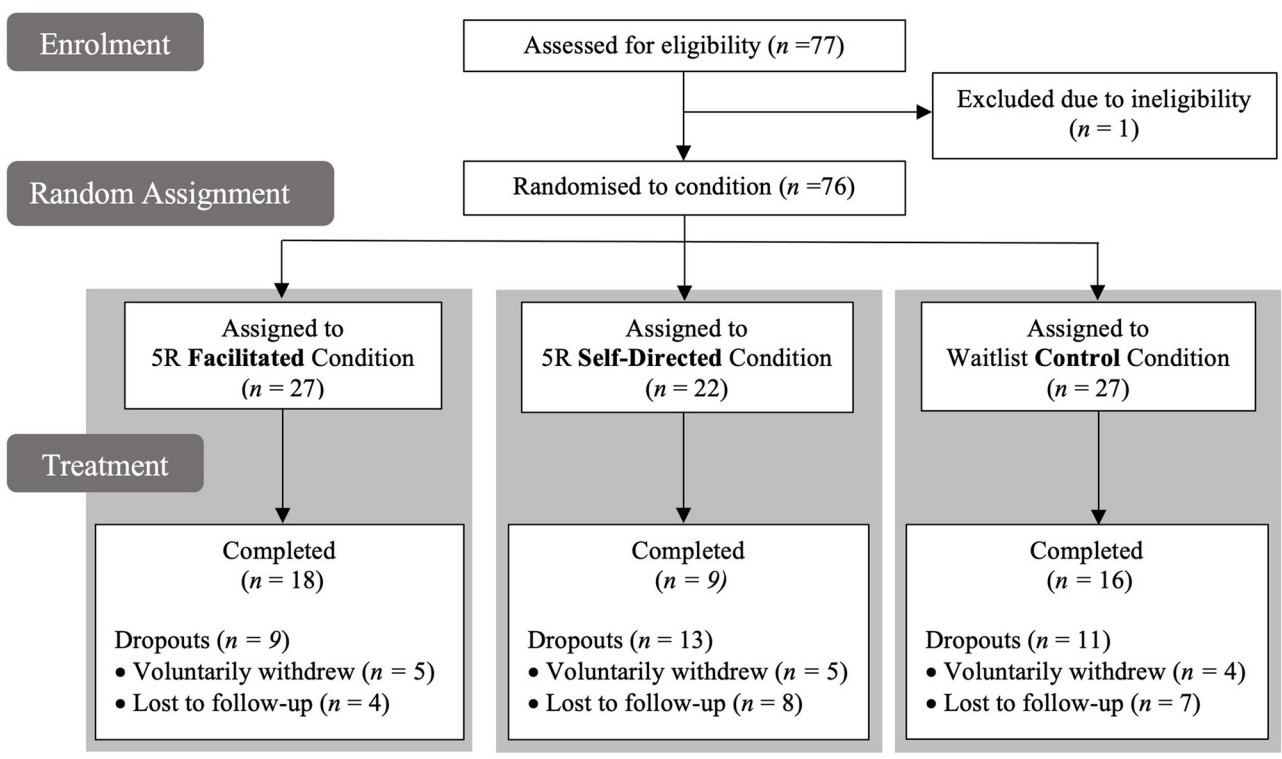

**Fig 1. Consort diagram for the study.**

Eligible participants were assigned to 12 cohorts, each containing up to eight leaders. Participation in the study was overseen by eight postgraduate organizational psychology students who worked in pairs to manage each cohort and whose involvement in the project was the concluding part of their formal training for a Master's degree in Organizational Psychology. Following the procedure recommended by Kernan et al. [67] and Suresh [68], a stratified block randomization process was used to assign cohorts to one of three experimental conditions—either 5R *Facilitated* or 5R *Self-directed* (the two *Treatment* conditions) or a waitlist *Control*. Stratification also ensured that each pair of psychologists ran one cohort in each condition. As can be seen from the consort diagram presented in Fig 1, follow-up data were obtained from 18 (67%) of the 27 participants who completed the facilitated version of 5R, from 9 (41%) of the 22 who completed the self-directed version, and from 16 (59%) of the 27 assigned to the control condition. Yet despite some indication of uneven completion, this did not vary significantly across conditions (due either to not completing the T2 survey, $\chi^2(2) = 3.40$, $p = .18$, or not completing the program: $\chi^2(2) = 1.26$, $p = .53$).

## Procedure

Prior to the start of the study, the postgraduate organizational psychology students who were going to facilitate 5R during the trial took part in approximately 20 hours of training (conducted by three of the authors) to learn about various aspects of the program and its delivery. Before the program started, all participants completed a pre-test (T1) survey and those in both treatment conditions took part in an online induction session that was facilitated by these students and explained the structure and requirements of the program. Participants in the treatment conditions then progressed through the five program modules, spaced two weeks apart.

Those assigned to the waitlist control condition were told that they would start the 5R program in two months' time and completed their T2 survey before doing so.

The 5R program was comprised of on-line content that participants worked through in a sequence of five modules that each took about an hour to complete. To render it suitable for online delivery, the content of the modules was substantially modified from the original version of 5R (described in detail in [43]). Amongst other things, each included (a) 10-minute videos designed to summarize key points (for an example see: https://www.youtube.com/watch?v=aPEm1qwSejI), (b) a hospital-based case study that illustrated significant ideas, and (c) ILI-related exercises that provided participants with an opportunity to reflect on their own and others' identity leadership experiences. Departing from the previous face-to-face version of 5R, the fifth online module was adapted to focus more on the challenges of sustaining healthy team functioning over time and was subsequently re-titled *Reinforcing* (rather than Reporting).

Each module involved participants reading online material, watching short videos, engaging in a range of interactive exercises that included self-reflective elements, and also completing self-managed activities in between modules, either on their own or with their team members. The modules also engaged with the content of short articles that summarized research related to the theme of the module, and which participants could read in their own time (e.g., see [69, 70]). The overall structure of the resulting program is summarized in Fig 2 and a process map of the program is presented in Fig 3.

After completing each online module and associated readings, participants conducted relevant activities with their teams (e.g., social identity mapping, participative goal setting). The purpose of these activities was to encourage participants to engage directly with the groups they need to lead. After each module, participants in the facilitated condition then also took part in a 90-minute facilitated on-line workshop. The purpose of these workshops was to engage closely with the logic of identity leadership and to bring this to life through discussion

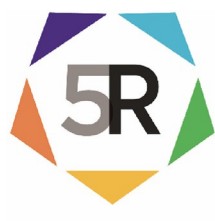

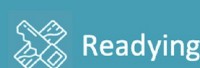 *Why does 'we' matter?* Understanding the value of groups for leadership and of ways to harness this.

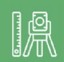 Reflecting — *Who are we?* Using social identity mapping to identify followers' important group memberships and areas for growth.

Representing — *What are we about and what do we want to be?* Clarifying group values, norms and aspirations.

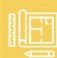 Realizing — *How do we become what we want to be?* Developing strategies to achieve group goals and embed group identity.

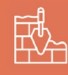 Reinforcing — *How can we be better?* Ensuring that identity leadership is ethical, healthy and sustained.

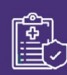

**Fig 2. The structure of the online 5R program.**

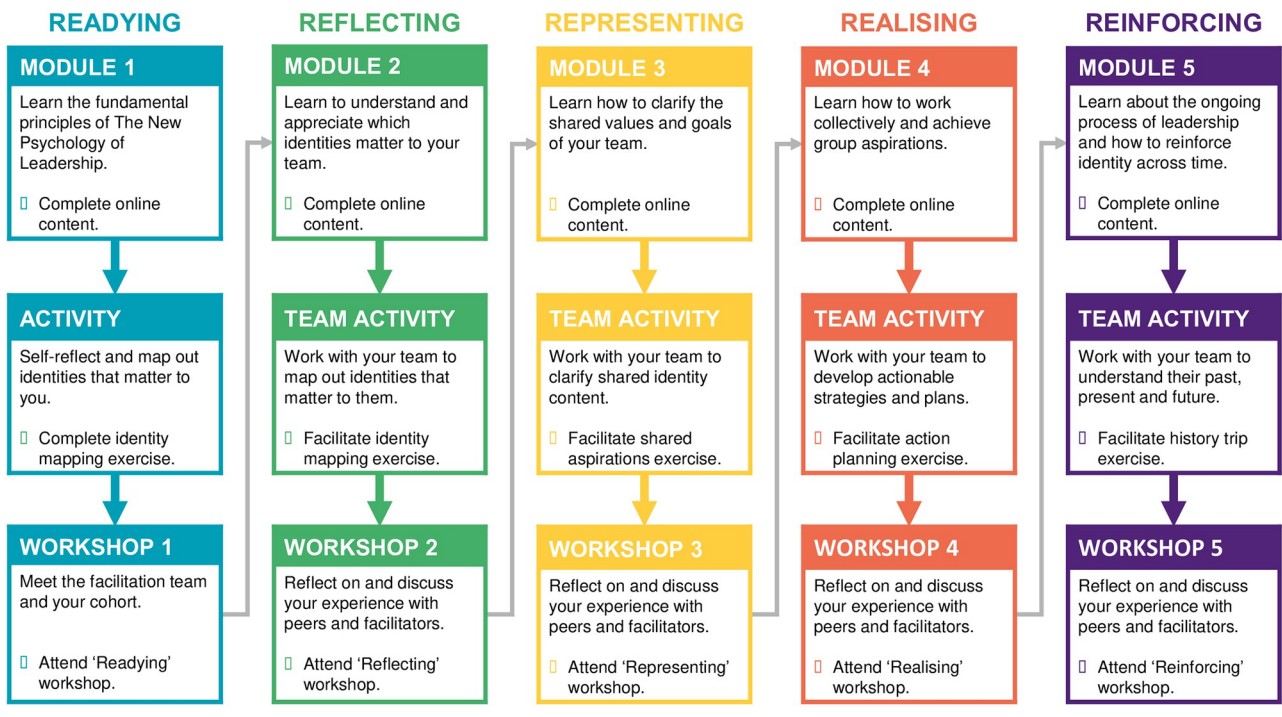

**Fig 3. Process map of participants' progress through the 5R program.**

of the online module content, the readings, the case study and the activity they had conducted with their teams. Participants in the self-directed condition worked through the online modules, completed the readings, and conducted the related activities with their teams, but they did not participate in the group workshops.

At the end of the program, all participants completed the T2 survey. Those in the treatment conditions (both facilitated and self-directed) were also invited to undertake one-on-one interviews with the facilitators to discuss their experience of the program. If they preferred, they were invited to provide comments on the program and their experience via e-mail.

## Measures

Details of the measures included in the T1 and T2 surveys are provided in Table 1. These included a manipulation check designed to assess whether participants had acquired knowledge of identity leadership related to their ability to lead their team as well as a control measure of leader identity pursuit (both relevant to H1), together with measures of multiple aspects of team functioning which previous research suggested might be positively impacted by identity leadership (relevant to H2). Participants also provided demographic details and were reassured that their responses were anonymous and confidential. The surveys included no other measures. Data from the study is available at https://osf.io/e82bd/?view_only=2d846d686e844a5e bd28347fbed32396.

## Results

### Tests of H1

Following the procedure of Haslam et al. [43], H1 was tested using a repeated-measures analysis of variance (AVOVA) examining knowledge of identity leadership at T1 and T2 as a

**Table 1. Measures included in the study.**

| Construct | Scale | No. items | Sample item | Scale endpoints |
|---|---|---|---|---|
| *Manipulation check* | | | | |
| Identity leadership knowledge | Steffens et al. [3] | 3 | *I know how to create a sense of cohesion within this team* | *1 = strongly disagree, 7 = strongly agree* |
| *Control measure* | | | | |
| Leader identity pursuit | Haslam et al. [43] | 4 | *I strive to create a sense among the people in my team that I am their leader* | *1 = strongly disagree, 7 = strongly agree* |
| *Measures of team functioning* | | | | |
| Team identification | Postmes et al. [71] | 1 | *I identify with this team* | *1 = strongly disagree, 7 = strongly agree* |
| Organizational identification | Postmes et al. [71] | 1 | *I identify with this organization* | *1 = strongly disagree, 7 = strongly agree* |
| Team goal clarity | Peters et al. [36] | 3 | *I have a clear sense of the goals of my team* | *1 = strongly disagree, 7 = strongly agree* |
| Organizational goal clarity | Peters et al. [36] | 3 | *I have a clear sense of the goals of my organization* | *1 = strongly disagree, 7 strongly agree* |
| Team reflexivity | Widmer et al. [72] | 4 | *As a team we usually take well-considered decisions* | *1 = strongly disagree, 7 = strongly agree* |
| Team psychological safety | Edmondson [73] | 3 | *Members of this team are able to bring up problems and tough issues* | *1 = very inaccurate, 7 = very accurate* |
| Inclusive team climate | Nishii [74] | 4 | *In my team everyone's input is actively sought* | *1 = strongly disagree, 7 = strongly agree* |
| Organizational citizenship | Lee & Allen [75] | 4 | *I give up time to help others who have work or non-work problems* | *1 = strongly disagree, 7 = strongly agree* |
| Work engagement | Schaufeli et al. [76] | 9 | *I am enthusiastic about my job* | *1 = never, 7 = always* |
| Team efficacy | Bohn [77] | 5 | *In my team we coordinate our efforts to complete difficult projects* | *1 = never, 7 = always* |

function of condition—comparing (a) treatment vs. control conditions and then (b) facilitated vs. self-directed conditions. As with all the other tests reported below, tests of H1 were based on the analysis of data from the 43 leaders who completed both T1 and T2 surveys. Descriptive and inferential statistics are presented in Table 2.

Supporting H1a, this analysis revealed a significant interaction between Time and Condition, $F(2,41) = 10.19$, $p < .001$, $\eta^2 p = 0.20$. Follow-up $t$-tests indicated that participants reported an increase in knowledge of identity leadership in the treatment conditions ($M_{T1} = 5.59$, $M_{T2} = 6.19$, $t(26) = 4.11$, $p < .001$, $d = 0.78$, a large effect) but not in the control condition ($M_{T1} = 5.80$, $M_{T2} = 5.65$, $t(15) = 0.81$, $p = .431$, $d = 0.20$, a small effect in the opposite direction). However, there was no evidence to support H1b in so far as ANOVA comparing facilitated and self-directed conditions revealed only a main effect of Time, $F(1,25) = 14.57$, $p < .001$, $\eta^2 p = 0.37$. This reflected the fact that knowledge of identity leadership increased in *both* the facilitated condition ($M_{T1} = 5.57$, $M_{T2} = 6.17$, $t(17) = 3.10$, $p = .003$, $d = 0.73$, a large effect) and the self-directed condition ($M_{T1} = 5.64$, $M_{T2} = 6.25$, $t(8) = 2.72$, $p = .026$, $d = 0.91$, also a large effect).

Similar analysis for leader identity pursuit (the control variable) also revealed a significant interaction between Time and Condition, $F(2,41) = 6.54$, $p = .004$, $\eta^2 p = 0.22$. Follow-up $t$-tests indicated that participants reported a *decreased* desire to pursue leader identity in the treatment conditions ($M_{T1} = 4.67$, $M_{T2} = 3.62$, $t(26) = 3.14$, $p = .004$, $d = 0.59$, a medium-sized effect) but no change in this motivation over time in the control condition ($M_{T1} = 4.88$, $M_{T2} = 4.69$, $t(15) = 1.29$, $p = .216$, $d = 0.32$, a small effect). Further analysis comparing facilitated and

**Table 2. Descriptive statistics and interaction F-values as a function of experimental condition (Facilitated + Self-Directed vs. Control).**

| Variable | Facilitated (N = 18) | | Self-Directed (N = 9) | | Control (N = 16) | | Time x Condition F-values | | |
|---|---|---|---|---|---|---|---|---|---|
| | T1 | T2 | T1 | T2 | T1 | T2 | 3 Conditions | (a) F/S-D vs. C | (b) F vs. S-D |
| Identity leadership knowledge | 5.57 (0.64) | 6.17 (0.76) | 5.64 (0.88) | 6.25 (0.48) | 5.80 (0.67) | 5.65 (0.56) | 4.97* | 10.19** | 0.00 |
| Leader identity pursuit | 4.76 (1.33) | 3.19 (1.68) | 4.47 (1.26) | 4.47 (1.25) | 4.88 (1.22) | 4.69 (1.28) | 6.53** | 3.67# | 5.88* |
| Teamfulness τ | 5.84 (0.58) | 6.18 (0.53) | 5.76 (0.66) | 6.14 (0.46) | 5.91 (0.40) | 5.92 (0.52) | 3.49* | 6.95* | 0.13 |
| Organizational alignment Ω | 5.91 (0.49) | 5.84 (0.71) | 6.06 (0.49) | 6.07 (0.43) | 5.74 (0.87) | 5.75 (0.83) | 0.03 | 0.09 | 0.73 |
| Team engagement ε | 5.82 (0.69) | 6.00 (0.60) | 5.76 (0.54) | 6.04 (0.51) | 5.82 (0.37) | 5.76 (0.36) | 3.42* | 6.38* | 0.37 |

*Note*:

** $p < .01$;

* $p < .05$;

# $p < .10$

τ Teamfulness = team reflexivity + team psychological safety + team goal clarity + inclusive team climate

Ø Organizational alignment = organizational goal clarity + organizational identification

ε Engagement = team identification + team organizational citizenship + team efficacy + work engagement

self-directed conditions also revealed a main effect of Time, $F(1,25) = 5.88$, $p = .023$, $\eta^2 p = 0.19$, and an interaction between Time and Condition, $F(1,25) = 5.88$, $p = .023$, $\eta^2 p = 0.19$. This reflected the fact that the decreased desire to pursue leader identity was apparent in the facilitated condition ($M_{T1} = 4.76$, $M_{T2} = 3.19$, $t(17) = 3.84$, $p = .001$, $d = 0.91$, a large effect), but not in the self-directed condition ($M_{T1} = 4.47$, $M_{T2} = 4.47$, $t(8) = 0.00$, $p = 1.00$, $d = 0.00$).

## Tests of H2

Given the large number of measures pertaining to H2, to avoid inflating Type I error, we initially explored opportunities for data reduction by conducting Principal Components Analysis of the 10 measures of team functioning. This analysis identified three factors with very similar structure at both Time 1 and Time 2. The small sample size precluded use of the emergent factor scores [78, 79] but because the factors made sense on conceptual grounds, we created compound measures of these by averaging participants' scores on each set of measures (as recommended by Kim and Mueller [80]).

The first construct was comprised of measures of team reflexivity, team psychological safety, team goal clarity, and inclusive team climate ($\alpha_{T1} = .82$; $\alpha_{T2} = .79$). Reflecting the fact that all of these measures related to leaders' sense of the healthy functioning of their teams, we refer to this factor as *teamfulness* (τ; akin to Weick & Roberts' [81] notion of collective mindfulness; see also [82]). The second construct, which we refer to as *organizational alignment* (Ω) was comprised of measures of organizational goal clarity and organizational identification ($r_{T1} = .46$; $r_{T2} = .74$). The third construct, which we refer to as *team engagement* (ε) was comprised of measures of team identification, work engagement, team efficacy and organizational citizenship ($\alpha_{T1} = .67$; $\alpha_{T2} = .71$). Descriptive and inferential statistics for the three constructs are presented in Table 2.

To test H2 we then conducted separate ANOVAs on scores on these composite measures. As can be seen from Table 2, consistent with H2a, analysis of teamfulness scores (τ) revealed a significant interaction between Time and Condition, $F(2,41) = 3.36$, $p = .045$, $\eta^2 p = 0.08$. Follow-up t-tests indicated that this arose from the fact that there was an increase in teamfulness in the treatment conditions ($M_{T1} = 5.80$, $M_{T2} = 6.16$, $t(26) = 3.87$, $p < .001$, $d = 0.73$, a large

effect) but not in the control condition ($M_{T1}$ = 5.91, $M_{T2}$ = 5.92, $t$(15) = 0.11, $p$ = .914, $d$ = 0.03). However, there was no evidence to support H2b in so far as repeated-measures ANOVA comparing the facilitated and self-directed conditions revealed only a main effect of Time, $F$(1,25) = 13.76, $p$ = .001, $\eta^2$p = 0.34. This reflected the fact that teamfulness increased in *both* the facilitated condition ($M_{T1}$ = 5.84, $M_{T2}$ = 6.18, $t$(17) = 2.99, $p$ = .008, $d$ = 0.70, a medium-to-large effect size) and the self-directed condition ($M_{T1}$ = 5.76, $M_{T2}$ = 6.14, $t$(8) = 2.55, $p$ = .034, $d$ = 0.85, a large effect).

In contrast, no effects emerged from analysis of participants' organizational alignment scores ($\Omega$) comparing treatment and control as well as the two treatment conditions (all $F$-values < 1.00). However, again supporting H2a, analysis of team engagement scores ($\varepsilon$) revealed a significant interaction between Time and Condition, $F$(2,41) = 6.33, $p$ = .016, $\eta^2$p = 0.24. Follow-up $t$-tests indicated that this reflected the fact that there was an increase in team engagement in the treatment conditions ($M_{T1}$ = 5.80, $M_{T2}$ = 6.01, $t$(26) = 2.75, $p$ = .011, $d$ = 0.53, a medium-sized effect) but not in the control condition ($M_{T1}$ = 5.82, $M_{T2}$ = 5.76, $t$(15) = 1.34, $p$ = .200, $d$ = 0.34, a small effect in the opposite direction). However, there was again no evidence to support H2b in so far as repeated-measures ANOVA comparing facilitated and self-directed conditions revealed only a main effect of Time, $F$(1,25) = 7.63, $p$ = .011, $\eta^2$p = 0.23. Again this reflected the fact that team engagement increased in *both* the facilitated condition ($M_{T1}$ = 5.81, $M_{T2}$ = 6.00, $t$(17) = 1.60, $p$ = .129, $d$ = 0.38, a small-to-medium effect) and the self-directed condition ($M_{T1}$ = 5.76, $M_{T2}$ = 6.03, $t$(8) = 4.35, $p$ = .002, $d$ = 1.45, a large effect).

## Tests of H3

Given the patterns of results reported above, it only made sense to test whether H3 was supported in the case of teamfulness ($\tau$) and team engagement ($\varepsilon$) scores, as only here was there an effect that might be explained by participants' knowledge of identity leadership. To test whether this was the case, we conducted indirect effect analysis using bias-corrected bootstrapping with 5000 resamples (using PROCESS [83]). Supporting H3, analysis revealed a significant indirect path of condition on increased teamfulness through increased knowledge of identity leadership, IE = .20, SE = .09, 95% CIs = .05, .42. However, there was no equivalent indirect effect to team engagement, IE = .05, SE = .06, 95% CIs = -.06, .18.

## Intention to treat

The fact that a reasonably high proportion of participants failed to complete the T2 survey raises the question of whether support for hypotheses would still be apparent if analysis included data from all participants who were recruited into the study. To address this possibility, we conducted intention-to-treat analysis [84, 85] that included all the available data from participants and imputed missing data by substituting in either (a) the mean value of each measure for each condition (a relatively liberal test of robustness) or (b) the last observation carried forward (LOCF) for each participant (a relatively conservative test; White et al., 2012).

Focusing on the more conservative of these methods and the key comparisons from which support for our main hypotheses was derived, $t$-tests following LOCF imputation supported H1 in indicating that participants reported an increase in knowledge of identity leadership in the treatment conditions ($M_{T1}$ = 5.46, $M_{T2}$ = 5.80, $t$(48) = 3.65, $p$ < .001, $d$ = 0.52, a medium-sized effect) but not in the control condition ($M_{T1}$ = 5.91, $M_{T2}$ = 5.83, $t$(26) = 0.81, $p$ = .424, $d$ = 0.16). Likewise, $t$-tests following LOCF imputation supported H2a and H2b in indicating that there was an increase in leaders' teamfulness ($\tau$) and their team engagement ($\varepsilon$) in the 5R treatment conditions ($\tau$: $M_{T1}$ = 5.79, $M_{T2}$ = 5.99, $t$(49) = 3.55, $p$ < .001, $d$ = 0.50, a medium-

sized effect; $\varepsilon$: $M_{T1}$ = 5.67, $M_{T2}$ = 5.79, $t(49)$ = 2.61, $p$ = .012, $d$ = 0.37, a small-to-medium-sized effect) but not in the control condition ($\tau$: $M_{T1}$ = 5.92, $M_{T2}$ = 5.93, $t(26)$ = 0.11, $p$ = .911; $\varepsilon$: $M_{T1}$ = 5.78, $M_{T2}$ = 5.74, $t(26)$ = 1.33, $p$ = .196, $d$ = 0.26, a small effect in the opposite direction). These results suggest that our hypotheses were robust to problems associated with missing data.

## Qualitative data

Qualitative feedback on the program was obtained from 21 of the 27 participants in the treatment conditions (78%; 13 in the facilitated condition, 8 from the self-directed condition). From these it was apparent that participants' experiences were overwhelmingly positive. The following comments are typical of those provided by leaders in the Facilitated condition:

> "[The program was] not too high-level in terms of the academic speak, it was just right. The mix of videos, readings and activities was engaging. I also liked how different participants facilitated different sections of the workshop. It's important to take the time to reflect with the team".
>
> (KS, m, C1)

> "I believe the program was an excellent experience and a great way to re-calibrate some of the traditional training investments in boss-centric leadership, to truly focus on the followers."
>
> (DG, f, C4)

> "It has been great. I did get a lot of information out of the program that I can use moving forward to help me guide my team. The program is about things that l feel strongly about and will continue to be one of them. I have learned how to get the team engaged, and it is important to not dictate. Once you come down to their level, they will respect you".
>
> (TC, m, C11)

Broadly similar in its positivity, the following feedback is representative of that provided by participants in the self-directed condition:

> "I've found the conversations I've had . . . with my direct reports to be some of the most rewarding parts of the program. The reason for this is two-fold: (a) there is a sense of social identity in these groups and it is motivating me to perform and act in the best way for the groups and (b) the simple mantra of 'what gets checked gets done'."
>
> (MG, m, C7)

> "I really enjoyed the program and learnt a lot from it."
>
> (PM, f, C7)

> "I loved it. The relevance of the content was good. It was very engaging in the way it was delivered with the modules, the takeaways and the real-life examples. The time investment was good, not too long or too short. It was good. I really enjoyed it."
>
> (JG, m, C9)

At the same time, a number of participants had suggestions for ways in which the 5R program could be improved. Most of these focused on providing more opportunity for interaction

with other participants. However, as the following examples indicate, such feedback came almost entirely from participants in the self-directed condition:

> "My dissatisfaction is that I have a lot of loose ends in my thoughts and would love further conversation with others partaking in the program.. . . I am enjoying the program but feel restricted to only have access to my own interpretation."

> (PM, f, C7)

> "I thought that the program was lacking a little without interaction between participants and I found that a bit challenging.. . . I did find the content valuable but I think it could have been so much better if I could have discussed it with others."

> (GI, m, C9).

## Discussion

The goal of the present research was to explore the benefits of a leadership development program—5R—designed to build leaders' understanding of the importance of shared social identity for leadership and to help them develop the insight and skills needed to create, advance, represent and embed a sense of social identity ('us-ness') in the teams they lead. Where traditional approaches to leader(ship) training and development focus on leaders in isolation and often in contexts removed from their normal sphere of activity in ways that aim to build leader identity, 5R encourages leaders to engage directly with the groups they are attempting to lead and gives them the skills to do so with confidence in ways that build identity leadership. In this way, and in line with suggestions that leadership development should focus on the specific contexts in which leaders operate rather on leaders (and leader identity) in isolation, the program is designed to help leaders mobilize followers (the team members for whom leaders have responsibility) rather than to exclude them from the leadership process and the broader dynamics of organizational development and change.

These and other differences between 5R and traditional approaches to leadership are summarized in Table 3.

Although previous studies have pointed to the benefits of 5R, interpretation of their findings has been clouded by a number of factors. Primary amongst these is a lack of experimental

**Table 3. Some key differences between 5R and traditional approaches to leader(ship) development.**

| | Traditional approaches | 5R |
|---|---|---|
| Focus on developing | The leader as an individual | The leader as a group member |
| | Leader identity | Identity leadership |
| | Personal identity ('I') | Social identity ('we') |
| Implementation content | Primarily intellectual | Intellectual and practical |
| | Personal reflection | Collective reflection and engagement |
| | One-shot | Developmental |
| Implementation context | Away from the group being led | With the group being led |
| | Excludes followers | Includes followers |
| | Increases psychological distance from followers | Reduces psychological distance from followers |
| Impact on leaders | Increases personal awareness | Increases collective awareness and 'teamfulness' |
| | Increases sense of superiority and hubris | Does not increase sense of superiority and hubris |
| | Rewards and encourages narcissism | Does not reward or encourage narcissism |

control. The fact that previous trials have been delivered by 5R developers also raises some questions about the generalizability of program's benefits as the knowledge and motivation of these facilitators may not be shared by others. Finally, the fact that previous trials have all been conducted face-to-face raises questions about whether the program can be successfully adapted for on-line delivery. Indeed, this question has assumed broad relevance for the leadership development industry as a result of increased demand for online-based training in the wake of COVID-19 [86, 87]—although in many ways the pandemic only accelerated developments of online-based (online-augmented or blended) training that were already underway [88, 89].

The results of this randomized controlled trial provide reasonably clear answers to these questions. In the first instance, supporting H1, participation in both facilitated and self-directed versions of 5R served to produce significant and large increases in participants' knowledge about leadership as a social identity process (e.g., in ways suggested by [1, 3, 4]). At the same time, while it increased their knowledge of identity leadership, participation did not increase participants' motivation to develop an identity as a leader per se (i.e., pursue a personal *leader identity*; [8, 9, 51, 53] as per Table 3). Indeed, a significant *reduction* in this motivation was observed among participants in the facilitated condition. These patterns are significant for two reasons. First, because, commensurate with the logic of non-equivalent dependent variable design [63, 64], they speak to targeted impact of the 5R program. In particular, they suggest that the changes it produced are not indiscriminate (e.g., relating to an uplift in all forms of leader motivation or to common method variance) but are related specifically to participants' desire to develop as leaders by engaging more effectively with their teams. Relatedly, second, they suggest that the program avoided cultivating a sense of superiority among leaders that might set them apart from followers and thereby compromise their capacity to lead (in ways discussed by [11, 90–92]).

More substantially, in line with H2, participation in both facilitated and self-directed versions of 5R also had an impact on leaders' perceptions of their teams and of those teams' functioning. To assess this, we administered a series of measures designed to capture a range of organizational and team processes that previous research had suggested might be positively impacted by a sense of shared social identity and the leadership that builds this (e.g., [15, 21, 30, 43]). Factor analysis supported conceptual aggregation of these measures into three distinct constellations. One of these was related to leaders' organizational alignment (their organizational identification and sense of organizational goal clarity) which was unaffected by the intervention. However, participation in the 5R program had a positive and moderate-to-large impact on the other two constructs—both of which related more directly to team functioning.

The first of these constructs, which we termed team engagement, was associated with leaders engaging more closely with their teams, as captured by measures of their team identification, their team-directed organizational citizenship, their sense of team efficacy, and their work engagement. The second factor was associated with leaders feeling more secure and assured in their teams as captured by measures of team reflexivity, team psychological safety, team goal clarity, and inclusive team climate. This construct, which we have referred to as *teamfulness*, aligns closely with the sense of collective mind(fulness) that Weick and colleagues identify as critical to the success of high-reliability organizations—primarily because it underpins the heedful interrelating between leaders and team members that allows those teams to negotiate organizational complexity with confidence [81, 82] (see also [93] for a recent discussion). More generally, it is clear that this construct is theoretically aligned with the logic of social identity theorizing (and the aims of 5R), in so far as this sees the internalization of social identity as the process that makes co-ordinated team and organizational behavior possible [31, 94–96].

Support for our analysis of 5R's impact was also provided by regression modelling to establish whether the knowledge of identity leadership that was acquired through participation in 5R was implicated in leaders' increased team engagement and teamfulness. Supporting H3, this modelling showed that it was by reflecting on, and acquiring, skills of identity leadership that leaders felt secure and assured in their dealings with their team. At the same time, though, there was no evidence that knowledge of identity leadership was implicated in the increased team engagement that leaders reported. One obvious possibility is that this is because it was the 5R team activities themselves that had this effect. Indeed, this would accord with previous evidence that leadership training is effective because (and to the extent that) it encourages meaningful interaction with the teams that participants lead [46–50].

While there was general support for our experimental hypotheses, one unexpected finding was that the extent of this support did not vary as a function of the mode of 5R delivery. On the basis of previous research (e.g., [46]) we had expected that the benefits of the program would be more pronounced among those whose participation was facilitated rather than self-directed. However, as noted above, it was only on the control measure that assessed participants' pursuit of leader identity that any difference between these two conditions was observed. Insofar as we expected participation in 5R to reduce participants' leader identity pursuit (as per Table 3), this suggests that some of the lessons of the program were internalized more by participants in the facilitated version of the program than by those in the online version. Consistent with this, there was some evidence in participants' post-experimental qualitative feedback that those in the Facilitated condition were more engaged with the program and more enthusiastic about the experience of participating in it than their Self-Directed counterparts. In line with the conclusions of Lacerenza et al. [46], given a choice, this is therefore the version of the program that we would recommend (while noting that the Self-Directed version still appears to be beneficial). Going forward, it would also be worthwhile establishing whether the differences that were identified in qualitative feedback have more nuanced impact on program outcomes beyond those that were captured in the present study—for example, in affecting the likelihood of participants translating their learning from 5R into practice [97].

## Limitations and future research

As noted above, the primary motivation for the present study was to overcome the most significant limitations of previous trials of 5R. Nevertheless, like all research, it was not without limitations itself. Of these, the most obvious is that, in line with the logic of a Phase I clinical trial [98], the effects of the intervention were assessed relative to a no-treatment waiting-list control. A more stringent (Phase II) test would involve gauging the efficacy of the program against another 'standard' leadership program that reflects 'treatment as usual', or to conduct a (Phase III) test against an evidence-based best-practice intervention (Martin et al., 2021). Notwithstanding the fact that tests of this form are extremely rare in leadership research [58–60], there would clearly be value in progressing through these phases of testing in future research. In this regard too, GROUPS 4 HEALTH—5R's sister program which promotes health by building social identity in the clinical realm—provides a good template for a graded series of tests with precisely this structure [99–101].

While the sample of participants for this study was sufficiently large to test our hypotheses appropriately, there would also be value in conducting a more ambitious (Phase IV) multi-site trial of the 5R program along the lines of work on the GILD project which has sought to test and validate the ILI in organizations around the world [15, 25]. Amongst other things, such a trial could usefully assess the appropriateness of the program for leaders not only in different countries and cultures but also in different sectors (e.g., those that differ along the dimension

of individualism–collectivism [102]) as well as the impact of leaders' participation in the program on their teams (something that, for logistical reasons, we were unable to assess in this trial). In this context too there would be value in conducting organization-wide tests—with internal facilitators who are at arms' length from 5R researchers—to assess the impact of leaders' participation in the 5R program on a broad suite of organizational outcomes that can be objectively assessed (e.g., by indices of performance, turnover, and vitality).

Ideally too, future trials would collect data speaking to the efficacy of 5R not only from leaders but also from members of the teams that they lead, since, as we noted in the Introduction, it is ultimately the followership of others that is the proof of leadership. Doing so would also mean that evidence of 5R's efficacy would not be based exclusively on the self-reports of participants. In designing the present research it had been our original intention to collect data of this form, but this proved logistically too challenging (largely because it placed a very high administrative burden on the organizations from which the participants were recruited). Nevertheless, trials that address this limitation are currently underway.

## Concluding comment

In their ground-breaking reflections on the nature of effective leadership, Peter Drucker and Carl Weick both observed that for leaders to be able to rise to the challenges of leading complex teams they need to become one with those teams. *"The leaders who work most effectively, never say 'I'"*, wrote Drucker, *"And that's not because they have trained themselves not to say 'I'. They don't think 'I'. They think 'team'. They understand their job to be to make the team function. . .. There is an identification with the task and with the group"* ([103], p. 14]; cited in [1]). Likewise, Weick observed that highly reliable organizations are comprised of teams whose leadership has led them to *"act as if they are a group"* ([81], p.360]).

Not only, then, is social identity one of the key things that effective leadership builds, but so too leadership is one of the things that social identity makes possible (Reicher et al., 2005). Indeed, as Drucker and Weick suggest, it is the capacity to channel and mobilize the power of social identity that ultimately supports and bears testament to any individual's leadership [12]. Accordingly, the principal value of a leadership development program such as 5R is that it affords leaders the opportunity to understand the importance of social identity for team functioning and then to engage in activities that help to build it. In short, it helps to create engaged and *teamful* leaders who are mindful of the need to make 'us' both a psychological and a material force in the world—and who then have the confidence and skills to do so.

## Acknowledgments

We are grateful to Jack Chapman, Sage Cullen, Joanna Goh, Wenxuan Jiang, Brittany Powell, Jordan Schaefer, Melpomene Tantalos, and Elodie Tischer for their help delivering the trial of the 5R Leadership Development program reported in this paper.

## Author Contributions

**Conceptualization:** S. Alexander Haslam, Jordan Reutas, Sarah V. Bentley, Blake McMillan, Kim Peters, Niklas K. Steffens.

**Data curation:** S. Alexander Haslam, Jordan Reutas, Niklas K. Steffens.

**Formal analysis:** S. Alexander Haslam, Jordan Reutas, Kim Peters, Niklas K. Steffens.

**Investigation:** S. Alexander Haslam, Jordan Reutas, Sarah V. Bentley, Blake McMillan, Madison Lindfield, Mischel Luong.

**Methodology:** S. Alexander Haslam, Jordan Reutas, Sarah V. Bentley, Mischel Luong, Kim Peters, Niklas K. Steffens.

**Project administration:** S. Alexander Haslam, Jordan Reutas, Sarah V. Bentley, Blake McMillan, Madison Lindfield, Mischel Luong.

**Resources:** Madison Lindfield, Kim Peters.

**Supervision:** S. Alexander Haslam, Jordan Reutas, Blake McMillan, Niklas K. Steffens.

**Validation:** Kim Peters, Niklas K. Steffens.

**Visualization:** Jordan Reutas.

**Writing – original draft:** S. Alexander Haslam.

**Writing – review & editing:** S. Alexander Haslam, Jordan Reutas, Sarah V. Bentley, Blake McMillan, Madison Lindfield, Mischel Luong, Kim Peters, Niklas K. Steffens.

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
