## [Decision Letter · Decision Letter 0]

20 Apr 2023

PONE-D-23-04909Building engaged and ‘teamful’ leaders: A randomised controlled trial of an identity leadership interventionPLOS ONE

Dear Dr. Haslam,

Thank you for submitting your manuscript to PLOS ONE. After careful consideration, we feel that it has merit but does not fully meet PLOS ONE’s publication criteria as it currently stands. Therefore, we invite you to submit a revised version of the manuscript that addresses the points raised during the review process. The manuscript has now received two reviews (comments below).  Please try and address all comments, if possible, especially focusing on the changes required by Reviewer 1 in relation to developing a more theoretical rationale for the study and one which provides a stronger narrative.  

We look forward to receiving your revised manuscript.

Kind regards,

Charlotte Lennox

Academic Editor

PLOS ONE

“I have read the journal's policy and the authors of this manuscript have the following competing interests: Intellectual property pertaining to the 5R Leadership Development Program is asserted, owned, and licensed by the University of Queensland. The university and its agents have a commercial as well as an intellectual interest in this program. The fourth author is the Director of Identify Leadership which is formally licensed to deliver 5R.”

4. We note that you have referenced (Gopinathan, D. (2017). Exploring the robustness of the 5R leadership development program. Unpublished Masters thesis, University of Queensland.) which has currently not yet been accepted for publication. Please remove this from your References and amend this to state in the body of your manuscript: (ie “Bewick et al. [Unpublished]”) as detailed online in our guide for authors

Reviewers' comments:

Reviewer's Responses to Questions

**Comments to the Author**

1. Is the manuscript technically sound, and do the data support the conclusions?

Reviewer #1: Partly

Reviewer #2: Yes

2. Has the statistical analysis been performed appropriately and rigorously? 

Reviewer #1: N/A

Reviewer #2: Yes

3. Have the authors made all data underlying the findings in their manuscript fully available?

Reviewer #1: Yes

Reviewer #2: No

4. Is the manuscript presented in an intelligible fashion and written in standard English?

Reviewer #1: Yes

Reviewer #2: Yes

5. Review Comments to the Author

Reviewer #1: In the current paper, the authors explore the effect of facilitated and learner self-directed versions of identity leadership intervention (5R program) on identity leadership knowledge and team emergent states of team engagement and ‘teamfulness.’ The authors draw on the social identity approach to leadership to explain leaders’ capacity to impact the groups they lead by facilitating a sense of social identity (or ‘us-ness’) that those leaders create, advance, represent, and embed for the group. I commend the authors for their effort to conduct a randomized control study with team leaders to test the research predictions. The question addressed by the paper is important theoretically and practically, with the potential to contribute to different literature streams.

Nonetheless, some theoretical and methodological issues and concerns should be addressed to develop this work further. My overall impression is that the paper's theoretical contribution could be clearer and better communicated. And that the study suffers from methodological issues that should be addressed or clarified by mitigating the interpretation of the results.

Major issues and concerns

1. Theory and hypothesis building (writing) –

(a) Research gap - the intro and later the theory development part present lengthily the social identity approach to leadership and why it is important or successful, reviewing finding on leadership prototypicality (central for theory, but not for this research) and the social identity approach to leadership (GILD), stressing the importance of the general approach. Then the inro moves to the research questions about the development of identity leadership, describing the 3R and then 5R program and the supportive findings from field studies in different contexts. Finally, on page 8, the authors present the gaps in previous research – lack of randomized design, concern for experimenter bias, and face-to-face delivery. It seems that all these are broadly framed as methodological limitations. As a reader, I was looking to understand if there is a theoretical gap that this research intends to address or if the aim of the research is limited to strengthening the robustness of the 5R program as an efficient tool for developing identity leadership. Understanding that this work is about leadership development within a particular leadership approach (social identity), I expected the intro to locate this work in a broader context. For example - how is this program framed within, and is there any contribution to, the literature on leadership development programs in general? Within this scope and general learning/training scope – how do the different versions of facilitated vs. Self-Directed contribute to this or the general literature?

(b) Contributions – It will be helpful to start with an intro that briefly highlights the general context (like the social identity approach to leadership, leaders' identity, and leadership development), the research gap, and the expected contribution to hook the reader to the paper. If there is a broader theoretical contribution (I think there might be – for leadership development in general, or the difference between leaders' identity and social identity, which I find interesting), it should be better communicated and spelled out. Even if the aim is more limited to addressing the previous empirical limitation to support the theory and findings, it should be clear and communicated early on.

(c) Hypotheses – the logic explaining the three hypotheses is very briefly put forward, and also I was a bit surprised by parts of it. The hypotheses, as I understand, are:

H1: the program would increase participants’ knowledge and skills of identity leadership.

H2: the program would increase leaders’ self-reported ability to work with their teams to address their collective challenges and goals.

H3: the relationship between participation in the program and leaders' reported ability to work with their teams to address their collective challenges and goals will be mediated by their knowledge and skills of identity leadership.

This issue of leaders' self-report caught me by surprise; I will address the methodological CMV issue separately. Theoretically, reading the paper, I expected that intervention would increase leaders' skills and impact team members and the team state and processes. If the hypotheses are about leaders' perceptions and beliefs in their ability, the framing and build-up of the hypotheses should lead to that.

2. Research methods – here, I have several concerns.

(a) Same source bias – although the data was collected at different points in time and there is an intervention, it is still the same source and the same criteria, which is a significant concern for common method variance. You can see Chang et al. (2010) and Podsakoff et al. (2003) for more info on the problem and possible remedies.

(b). Constructs & measures – the items of team emergent states constructs, such as psychological safety, work engagement, etc., are about the team, but the report is of the leader. The best option would have been to collect data from team members and aggregate it to the team level. Using the current data set, some of the items are not about the team but a personal report (i.e., work engagement, "I am enthusiastic about my Job." Team identification, " I identify with this team"). This is a leader's report about himself, which is later aggregated to a construct called "Team engagement" or the "teamfulness." Looks like an operationalization gap between the items and the construct. This is somewhat confusing and poorly aligned with the hypotheses about leaders' perception of their ability to address the team challenges.

(c) It is written in the limitation and is indeed a limitation that the control had no treatment. This concern is amplified given that it is all a self-report.

Chang, S. J., Van Witteloostuijn, A., & Eden, L. (2010). From the editors: Common method variance in international business research. Journal of International Business Studies, 41(2), 178–184. https://doi.org/10.1057/jibs.2009.88

Podsakoff, P. M., MacKenzie, S. B., Lee, J. Y., & Podsakoff, N. P. 2003. Common method biases in behavioral research: A critical review of the literature and recommended remedies. Journal of Applied Psychology, 88(5): 879–903.

3. Results and analysis – the results and analysis are statistically significant, with a small sample size which suggests a strong effect size (can you provide effect size?). My main concern is not with the statistical analysis in this part but that it is not aligned with the theoretical constructs (previous point 2b).

4. Discussion – it seems like this section is underdeveloped. It is mainly an elaborated summary of the findings. This is partially related to the points I raised in the intro. If the gaps in the literature can be more clearly articulated in the introduction, then the discussion and contribution can be more developed.

Minor issues and suggestions for the authors.

A minor point in the result you report on H3a (p. 21); while there was no such hypothesis before – there was only a general hypothesis (H3).

To conclude, my impression is that this manuscript has been through several revisions and changes, and in its current form, not all its parts are well aligned, telling a compelling and clear story. More conceptual development and clarity, alongside rigorous empirical testing, are needed to substantiate and increase the present work's theoretical contribution and empirical robustness. I hope my review will not discourage the authors from further developing their research endeavor, and hopefully, they'll find my comments helpful as they continue their research.

Reviewer #2: I very much liked reading this paper and think it presents a very nice study on an important topic, namely a randomized control trial of an identity leadership intervention. The sample size is relatively small but the authors provided a power analysis which justifies the sample size. The analyses are all state of the art and I have nothing to add in this respect. The manuscript is well written and the hypotheses are well developed. I only have a few comments that may help to improve an already strong paper:

Major:

- I suggest downplaying the fact that the trainers/facilitators were not the authors themselves as the graduate students performing these roles were themselves trained by the authors and so one could say that they indirectly come from the same school of thought.

- in the sbtract, I would like to see a little more detail on the study design, Ns and results

- I like the fact that the training reduced the agreement to the leader identity pursuit item and I think you correctly identify this as evidence for the training reducing feelings of superiority among participants. This is well explained on p. 24 but then on p. 26 where you discuss the fact that this was only the case in the facilitated and not the self-directed condition you are a bit short - I think it would be good to elaborate this difference.

- In the limitations section I would add the fact that you have not measured any effects on the leaders' followers which would be good to indicate as a desiderate for future work.

Minor:

- p. 3: I would suggest replacing the this in the phrase "...causal status of this as a determinant..." by something more meaningful (e.g., prototypicality, or "being one of us")

- p3, last line "Develop" should be "Development"

- p. 4, line 3: I would suggest adding the reference Bracht et al. - Bracht et al. have just published (2023, Applied Psychology) a paper that directly speaks to the ILI robustness

- p. 27, second last line: You refer to an "in press" paper that is not listed in the references

6. PLOS authors have the option to publish the peer review history of their article (what does this mean?). If published, this will include your full peer review and any attached files.

Reviewer #1: No

Reviewer #2: No

---

## [Author Response · Author response to Decision Letter 0]

28 Apr 2023

Key Editorial points

E•1. Developing a more theoretical rationale for the study and one which provides a stronger narrative. 

The key points that you and the reviewers raised related to the need to tighten the theoretical rationale for the study. This was something we were happy to do, having previously cut some discussion of the theoretical rational in order ensure the paper was not overly long. Specifically, we have elaborated on the rationale for the study in a number of places in the Introduction. In particular, on p.6 we now note that:

This approach differs from prevailing approaches to leader development which focus largely on improving the skills and mindsets of individual leaders. Work which has evaluated such programs indicates that they generally succeed in developing a person’s leader identity (their sense of themselves as a leader). In contrast to this focus on ‘me’ and ‘I’, the social identity approach embraces the idea that leadership is fundamentally a group process requiring leaders to look outwards towards their teams and to develop sense of ‘we’ and ‘us’.

E•2. Additional files

In your letter you also asked us to include (a) A marked-up copy of your manuscript that highlights changes made to the original version (labeled 'Revised Manuscript with Track Changes') and (b) An unmarked version of your revised paper without tracked changes. You said we should upload this as a separate file labeled 'Manuscript'. We have done both these things.

E•3. Style requirements

We have followed your instructions and ensured that the manuscript follows the journal’s style requirements — especially by reformatting references and using US spelling throughout.

E•4. Conflict of interest

We confirm that our stated conflict of interest does not alter our adherence to all PLoS ONE policies on sharing data and materials.

E•5. Details of data repository

We have provided an anonymized link in the paper to the data which has now been loaded onto the platform of the Open Science Foundation. This is available here: https://osf.io/e82bd/?view_only=2d846d686e844a5ebd28347fbed32396

E•6. Removal of reference

We have removed reference to the unpublished manuscript by Gopinathan (2017).

Specific points raised by reviewers

Reviewer 1

General comment. We very much appreciate the trouble that this reviewer had gone to in reviewing the paper. Clearly an expert in the field, their observations seemed appropriate and well judged. We appreciate the general statement that “The question addressed by the paper is important theoretically and practically, with the potential to contribute to different literature streams” and the reviewer’s sense that the paper is well-written and has something interesting to say, but agree that the paper could be sharpened by reflection on the range of points they make. These were as follows:

R1•1 Clearer discussion of the theoretical framework that informs the paper

The reviewer asked us to “locate [our] work in a broader context” of work on leadership and leadership development programs in a way that makes it clear what the distinctive theoretical of the paper (and the 5R program) is. 

The reviewer’s points here are very well taken, and in revising the manuscript we have tried to engage closely with this issue, in particular by spelling out the differences between our ‘we-focused’ approach and the ‘I-focus’ on programs that focus on developing leader identity. 

R1•2a The issue of common method variance (CMV)

The reviewer notes that “although the data was collected at different points in time and there is an intervention, it is still the same source and the same criteria, which is a significant concern for common method variance”. It is certainly the case that the study’s reliance on self-report data was a limitation (as we note in the Discussion). As we note in the text though, (on p.9) this was an issue we attempted to address by including a measure of participants’ “leader identity pursuit” as a control variable that has superficial similarity to our target dependent variable (identity leadership knowledge), but which taps different theoretical processes (as discussed by Haslam et al., 2022). In line with the logic of non-equivalent dependent variable design (Cook et al., 1979; see also Frese et al., 2003), this allows us to rule out the possibility that any changes we observe arise from non-specific responses to the intervention such as CMV (Podsakoff et al., 2012). In revising the paper in both the Introduction (p.9) and the Discussion (p.24) we have now made an explicit reference to the fact that this strategy was designed to address this problem.

R1•2b Construct measures

The reviewer makes the point that a lot of our measures related to leaders’ perceptions of the team (e.g., team psychological safety, team engagement). We agree that this is somewhat unorthodox; however, in line with the goal of 5R — which as we note in the Introduction was to encourage leaders to focus out on their team and to engage with it — we think that these measures make sense in so far as they capture the impact of the leader (and their participation in 5R) on team function. Indeed, in line with the idea that participation in 5R would make leaders mindful of their team (i.e., ‘teamful’), we think that it is important to capture the extent to which the program was perceived by leaders to have these effects — which the results suggest that it did.

R1•3 Inclusion of effects sizes

The reviewer points out that because we had a relatively small sample size but the effects were still statistically significant, this suggests that the effect sizes were quite large, and that it would be good to report them in the paper. This was a very good observation, and, as the reviewer recommends, we now report effect sizes for all effects. As the reviewer anticipated, most of the effects in our treatment conditions were of moderate-to-large size, whereas those in the control condition tended to be small (and sometimes in the opposite direction). 

R1•4 Extended Discussion

Related to R1•1, the reviewer observes that the Discussion was a bit underdeveloped and could do with being fleshed out to address the significance of the paper for the broader leadership literature. Again, this makes a lot of sense, and we have attempted to develop the Discussion along the lines they suggest. In particular, we have added this text at the start of the Discussion as well as a table (Table 3) that summarizes some of the key differences between 5R and traditional approaches to leadership:

Where traditional approaches to leader(ship) training and development focus on leaders in isolation and often in contexts removed from their normal sphere of activity in ways that aim to build leader identity, 5R encourages leaders to engage directly with the groups they are attempting to lead and gives them the skills to do so with confidence in ways that build identity leadership. In this way, and in line with suggestions that leadership development should focus on the specific contexts in which leaders operate, the program is designed to help leaders mobilize followers (the team members for whom leaders have responsibility) rather than to exclude them from the leadership process and the broader dynamics of organizational development and change.

Reviewer 2

General comment. This reviewer is also clearly very knowledgeable about social identity and the field of leadership and raises a range of important points in the process of providing very valuable feedback. We appreciate their general judgment that “I very much liked reading this paper and think it presents a very nice study on an important topic, namely a randomized control trial of an identity leadership intervention.” as well as their statement that “The manuscript is well written and the hypotheses are well developed”. Like Reviewer 1, though, s/he makes a series of good points that needed to be addressed in the revision. These were as follows:

R2•1. Downplay independence of instructors

The reviewer that we suggest downplay the independence of the trainers/facilitators as the graduate students performing these roles were themselves trained by the authors. This is a fair point, and we have followed the reviewer’s suggestion and downplayed this in both the abstract and the manuscript as a whole. 

R2•2. Provide more detail about the study in the abstract

The reviewer asked us to provide a bit more detail about the study’s design in the abstract. As they suggested, we have included the Ns for each condition and also provided more details about the study’s findings. 

R2•3. Clarification of results on leader identity pursuit measure

The reviewer asks us to clarify our discussion of the results on the control measure (leader identity pursuit) by explaining how the findings differed across the two treatment conditions. As they suggest, we have done this on p.26 by noting that insofar as we expected participation in 5R to reduce participants’ leader identity pursuit, the fact that this was seen in the Facilitated (but not the Self-Directed) condition suggests that some of the lessons of the program were internalised more by participants in the facilitated version of the program than by those in the self-directed version.

R2•4. Note that the absence of follower data is a limitation

As the reviewer (and R1) notes, we agree that ideally we would have gathered data from followers to assess the impact of leaders’ participation in 5R on them and the group as a whole. This had certainly been our intention, but it proved too logistically challenging. However, as we now note in the revised manuscript (on p.28), it is our intention to do this in future trials.

---

## [Decision Letter · Decision Letter 1]

12 May 2023

Developing engaged and ‘teamful’ leaders: A randomised controlled trial of the 5R identity leadership program

PONE-D-23-04909R1

Dear Dr. Haslam,

We’re pleased to inform you that your manuscript has been judged scientifically suitable for publication and will be formally accepted for publication once it meets all outstanding technical requirements.

Kind regards,

Charlotte Lennox

Academic Editor

PLOS ONE

Additional Editor Comments (optional):

Reviewers' comments:

Reviewer's Responses to Questions

**Comments to the Author**

1. If the authors have adequately addressed your comments raised in a previous round of review and you feel that this manuscript is now acceptable for publication, you may indicate that here to bypass the “Comments to the Author” section, enter your conflict of interest statement in the “Confidential to Editor” section, and submit your "Accept" recommendation.

Reviewer #1: All comments have been addressed

Reviewer #2: All comments have been addressed

2. Is the manuscript technically sound, and do the data support the conclusions?

Reviewer #1: Yes

Reviewer #2: Yes

3. Has the statistical analysis been performed appropriately and rigorously? 

Reviewer #1: Yes

Reviewer #2: Yes

4. Have the authors made all data underlying the findings in their manuscript fully available?

Reviewer #1: Yes

Reviewer #2: Yes

5. Is the manuscript presented in an intelligible fashion and written in standard English?

Reviewer #1: Yes

Reviewer #2: Yes

6. Review Comments to the Author

Reviewer #1: Nice work addressing the criticisms and situating the contribution within the existing literature. I think the intro and the additional table make the contribution of the work more apparent.

I now have only a minor suggestion about clarifying the construct measures, given the unorthodox way you measured constructs like – team identification, team goal clarity, etc. If the measures aim to "capture the extent to which the program was perceived by leaders to have these effects," as you explain in the letter, this should be communicated this way to the readers. Namely, I suggest explaining that these effects are on leaders' perception of these team constructs (i.e., 'teamful'); otherwise, it may be misleading, and the reader may be under the impression that it reflects changes in the team. This may be a path for future research to see if there is a later impact on the teams.

Reviewer #2: I commend the authors of addressing every point I raised to my satisfaction. They agreed that havoing follower data would have been good but I accept that this was not feasible in the present study. And I do not have any other concern.

7. PLOS authors have the option to publish the peer review history of their article (what does this mean?). If published, this will include your full peer review and any attached files.

Reviewer #1: No

Reviewer #2: No

---

## [Editor Report · Acceptance letter]

17 May 2023

PONE-D-23-04909R1 

Developing engaged and ‘teamful’ leaders: A randomized controlled trial of the 5R identity leadership program 

Dear Dr. Haslam:

I'm pleased to inform you that your manuscript has been deemed suitable for publication in PLOS ONE. Congratulations! Your manuscript is now with our production department. 

Kind regards, 

on behalf of

Dr. Charlotte Lennox 

Academic Editor

PLOS ONE